# Physiology of PNS axons relies on glycolytic metabolism in myelinating Schwann cells

**Marie Deck**[1]*, **Gerben Van Hameren**[1], **Graham Campbell**[1], **Nathalie Bernard-Marissal**[2], **Jérôme Devaux**[1], **Jade Berthelot**[1], **Alise Lattard**[1], **Jean-Jacques Médard**[3], **Benoît Gautier**[1], **Sophie Guelfi**[1], **Scarlette Abbou**[1], **Patrice Quintana**[1], **Juan Manuel Chao de la Barca**[4,5], **Pascal Reynier**[4,5], **Guy Lenaers**[5], **Roman Chrast**[3], **Nicolas Tricaud**[6]*

**1** INM, INSERM, Université de Montpellier, Montpellier, France, **2** Aix-Marseille University, INSERM, MMG, Marseille, France, **3** Departments of Clinical Neuroscience and Neuroscience, Karolinska Intitutet, Stockholm, Sweden, **4** Département de Biochimie et Génétique, Centre Hospitalier Universitaire, Angers, France, **5** Equipe Mitolab, MITOVASC, CNRS 6015, INSERM U1083, Université d'Angers, Angers, France, **6** I-Stem, UEVE/UPS U861, INSERM, AFM, Corbeil-Essonnes, France

* nicolas.tricaud@inserm.fr (NT); marie_deck@yahoo.fr (MD)

**Data Availability Statement:** All relevant data are within the manuscript and its Supporting Information files (supplementary information and raw data files).

## Abstract

While lactate shuttle theory states that glial cells metabolize glucose into lactate to shuttle it to neurons, how glial cells support axonal metabolism and function remains unclear. Lactate production is a common occurrence following anaerobic glycolysis in muscles. However, several other cell types, including some stem cells, activated macrophages and tumor cells, can produce lactate in presence of oxygen and cellular respiration, using Pyruvate Kinase 2 (PKM2) to divert pyruvate to lactate dehydrogenase. We show here that PKM2 is also upregulated in myelinating Schwann cells (mSC) of mature mouse sciatic nerve versus postnatal immature nerve. Deletion of this isoform in PLP-expressing cells in mice leads to a deficit of lactate in mSC and in peripheral nerves. While the structure of myelin sheath was preserved, mutant mice developed a peripheral neuropathy. Peripheral nerve axons of mutant mice failed to maintain lactate homeostasis upon activity, resulting in an impaired production of mitochondrial ATP. Action potential propagation was not altered but axonal mitochondria transport was slowed down, muscle axon terminals retracted and motor neurons displayed cellular stress. Additional reduction of lactate availability through dichloroacetate treatment, which diverts pyruvate to mitochondrial oxidative phosphorylation, further aggravated motor dysfunction in mutant mice. Thus, lactate production through PKM2 enzyme and aerobic glycolysis is essential in mSC for the long-term maintenance of peripheral nerve axon physiology and function.

## Introduction

Energetic metabolism is an essential parameter of neurons' function and survival [1–3]. Indeed, the nervous system consumes a large amount of glucose mainly to allow synapses to function but also organelles and vesicles transport along axons and action potentials firing [1].

**Funding:** European Research Council grant (FP7-IDEAS-ERC #311610), INSERM AVENIR, EpiGenMed Labex, The Neuromuscular Research Association Basel, Swedish StratNeuro program, Swedish Research Council grant (#2015-02394), and AFM-Téléthon (#20044).

**Competing interests:** The authors have declared that no competing interests exist.

In the peripheral nervous system (PNS), axons grow far from their cell bodies to reach distant targets such as muscles. Maintaining metabolic homeostasis on such a long distance is a real challenge for peripheral nerve axons [4].

To support neurons in this challenge, myelinating Schwann cells (mSC) cover the large motor axons with a myelin sheath. This insulating sheath, resulting from several turns of compacted plasma membrane, allows for ionic and electronic insulation forcing action potentials to "jump" from one node of Ranvier to the next, accelerating the nerve conduction velocity up to 120m/s [5]. Unfortunately, this insulation also hinders the diffusion of extracellular metabolites in the axonal environment mechanically limiting metabolic support to myelinated axons.

However, glial cells have also been shown to provide a direct trophic support to the axons they surround. Indeed, evidence exist for a metabolic coupling between glial cells and neurons in the CNS. Astrocytes and oligodendrocytes play a critical role in this process by metabolizing glucose into lactate and exporting it to the axon as a fuel for axonal mitochondria, a process known as the lactate shuttle [6–8]. Recently, mitochondrial respiration was shown to be dispensable for myelinating oligodendrocytes suggesting these cells can use glycolysis for ATP production and produce lactate [7, 9]. The disruption of monocarboxylate transporters (MCT), which mediate the traffic of metabolites, induced motor endplate dysfunction [10, 11], axonal damages and the degeneration of neurons *in vivo* [8, 12].

Lactate production in aerobic conditions has been well studied in cancer cells. Indeed, many type of cancer cells use aerobic glycolysis, also called the Warburg effect, to produce ATP as well as other metabolites required for cell survival and proliferation [13]. This metabolic shift to the aerobic glycolysis relies on the expression of Pyruvate Kinase M2 (PKM2) isoform instead of PKM1 [14]. PKM2 is a master regulator of glycolysis cumulating metabolic and non-metabolic functions as a protein kinase and transcriptional coactivator [15]. How PKM2 expression leads to more lactate is not clear. However, as PKM2 slows down pyruvate synthesis compared to PKM1, this may allow lactate dehydrogenase to metabolize pyruvate into lactate [15–17].

Recently, cases of reversible peripheral neuropathy have been observed following treatment with dichloroacetate (DCA), a chemical compound acting on lactate level in cells [18, 19]. Indeed, DCA is indirectly activating pyruvate dehydrogenase (PDH) and increasing the mitochondrial uptake of pyruvate, depleting cellular lactate. While the use of this compound to prevent cancer cells growth in several tumors is controversial [19, 20], it is also indicated to treat acute and chronic lactic acidosis and diabetes [21]. DCA side effects suggest the importance of lactate balance in the PNS function.

Here, we investigated the role of lactate in the physiology and function of axons and mSC of the PNS through the deletion of PKM2 in mSC (mSC–PKM2). Our results revealed the delicate equilibrium of lactate homeostasis in myelinated fibres of peripheral nerves and the critical trophic role of mSC in the support of axonal function in PNS.

## Material and methods

### Animals

Schwann cell specific ablation of PKM2 in adult mice (*Plp1-cre^ERT;PKM2^f/f*) was obtained by crossing *Plp1-cre^ERT* (JAX: 005975) [22] mice with *PKM2^f/f* (JAX: 024048) [14] to generate *Plp1-Cre^ERT;Pkm2^+/f* animals, which were in turn backcrossed with *PKM2^f/f* mice. *Plp1-cre^ERT* mouse line was chosen because it is the only commercially available line allowing selective and inducible recombination in myelinating Schwann cells. All transgenic lines were kept in C57/Bl6 background. Genotyping for all mutants was performed by PCR strategies using standard procedures and appropriate primers from Jackson Laboratories. Animals were kept under

French and EU regulations, following recommendations of the local ethics committee (#2019020908408066). Sacrifice was done after sedation with an overdose of $CO_2$.

**Tamoxifen administration.** The recombination of the floxed allele was induced by intra-peritoneal injection of tamoxifen (Sigma, T-5648) dissolved in corn oil (Sigma, C-8267) once every 24 hours for a total of 5 consecutive days, in 1 month animals at 180 µg/ gram mouse weight.

*DCA administration*: dichloroacetate (DCA, Sigma, D-54702) dissolve in water was administrated by gavaging six days per week during seven weeks, at 500 mg/ kg mouse weight.

## Behavioural studies

**Rotarod tests.** Locomotor coordination was performed with a rotating rod apparatus (Bioseb). Mice from 2 to 12 month were placed on the rotating rod and challenged using the following step: on day 1, mice was familiarized to the behavioral room and learned to stay on the Rotarod at constant speed of 8 rpm for 2 min. At day 2, locomotion performance was assessed on the accelerating Rotarod with a rate of 4 to 40 rpm for 2 min. Latency to fall was recorded on 3 trials separating by at least 15 min pauses. Data are expressed as the means from the 3 trials for each animal, normalized according to animal weight, +- standard error of the mean (SEM).

**Griptest.** Muscular strength for the 4 limbs was assessed with a grip test apparatus (Bioseb). Mice were held by the tail and allowed to grab the grid and then pulled backwards in a horizontal plane. The maximum force (measured in Newtons) applied to the grid was recorded. Data are expressed as the means from 3 trials for each animal, normalized according to animal weight, +- standard error of the mean (SEM).

**Motor nerve conduction velocity measurement.** Mice were anesthetized with 2% isoflurane and maintained at 37˚c on a hotplate. Left and right sciatic nerves were successively stimulated at the sciatic notch (proximal stimulation) and the ankle (distal stimulation) via a pair of steel needle electrodes (AD Instruments, MLA1302) with supramaximal pulses (7V) of 0.05 milliseconds delivered using a PowerLab 26T (AD Instruments ML4856). The distance between the 2 sites of stimulation was measured alongside the skin surface with fully extended legs. The latencies of the CMAP were recorded with a second pair of electrodes inserted between the digits of the hind paw and measured from the stimulus artefact to the onset of the M-waves deflection. NCV was calculated by dividing the distance between sciatic notch and ankle sites of stimulation by the subtraction of the distal latency from the proximal latency.

## Immunohistochemistry

For immunohistochemistry on cryosection, nerves were dissected and fixed 2hours in PFA4% at 4˚C. For spinal cord and DRG, mice were first perfused transcardially with PBS, and spinal cord and DRG were dissected and fixed in PFA4% overnight or 10min respectively. All tissues were then transferred to 30% sucrose overnight before embedding into Optimum Cutting Temperature (OCT, Tissue-Tek) medium. Cryostat sections of nerves or spinal cord (12 µm) were dried at room temperature (RT) for 15 min, washed in PBS, incubated 1h at RT in a blocking solution (0.25% Triton X-100 and 10% normal Goat serum in PBS) and incubated with primary antibodies in blocking solution overnight at 4˚C. For immunochemistry on teased fibers, nerves were isolated from mice, fixed for 10 min in Zamboni's fixative, washed in PBS and subsequently teased on glass slides. Slides were then dried overnight at RT and immunostained as described above. Primary antibodies: PKM1: 1/2500 (NBP2-14833, Novus Biologicals); PKM2 1/500 (SAB4200095, Sigma Aldrich); 2H3:1/80 (DSHB); Cleaved caspase3 1/400 (Cell Signaling); Neurofilament SMI32 1/1000; E-Cadherin 1/300 (MABT26, Merck); SV2 1/

100 (DSHB). Secondary donkey antibodies coupled to Alexa Fluor 488, Alexa Fluor 594, or Alexa Fluor 647 1/1000 (Invitrogen) and DAPI 1/1000 (Molecular Probes). Imaged were acquired with a Zeiss confocal microscope LSM710.

For neuromuscular junctions immunostaining, gastrocnemius muscles were dissected and fixed 20 minutes in PFA4% and incubated in 25% sucrose solution at 4˚C for 24 h. Tissues were embedded in OCT and stored at -80˚C before processing. Neuromuscular Junctions (NMJ) occupancy was quantified on 25-μm thick longitudinal gastrocnemius sections. Muscle sections were incubated overnight at 4˚C in blocking solution (2% BSA, 10% NGS, 0.1% Triton and PBS) with mouse anti-SV2 (Developmental Hybridoma Bank, 1/50). Sections were next incubated with goat anti-mouse IgG1 Cy3-conjugated antibody (Jackson ImmunoResearch, 1/500) and Bungarotoxin-488 (Molecular Probes, 1/500), and mounted in Mowiol mounting medium. NMJ were imaged on Axioplan fluorescence microscope (Zeiss) with a 20x objective.

Images that did show at least 5 Neurofilament positive cells were not included in the statistical analysis.

## Histology staining

Cresyl violet staining was undertaken on PFA perfused spinal cord. Cresyl violet solution was prepared with 0.1g cresyl violet acetate in 100ml distilled water and 250μl glacial acetic acid and filtered before use. Slides were kept at -20˚C and removed one hour before staining. Sections were rinsed 2 times in PBS for 5 minutes and for 1 minute in distilled water. Sections were put in pre-warmed (45 ˚C) cresyl violet solution for 20 minutes. Slides were then rinsed 2 times for 5 minutes in distilled water, followed by a 3 minute rinse in 90% ethanol and then 100% ethanol and cleared in xylene. Slides were mounted in DPX. Brightfield images were taken on a Zeiss Imager.D2 at 10 x magnification. Motor neurons were identified by morphology and size.

## Electron microscopy

Sciatic nerves were isolated from mice and immediately fixed in 2.5% glutaraldehyde and 4% PFA for 2h at RT, and postfixed in 2.5% glutaraldehyde in PHEM buffer (1X, pH 7.4) overnight at 4˚C. They were then rinsed in PHEM buffer and post-fixed in a 0.5% osmic acid for 2h at dark and room temperature. After two washs in PHEM buffer, the cells were dehydrated in a graded series of ethanol solutions (30–100%). The cells were embedded in EmBed 812 using an Automated Microwave Tissue Processor for Electronic Microscopy, Leica EM AMW. Thin sections (70 nm; Leica-Reichert Ultracut E) were collected at different levels of each block. These sections were counterstained with uranyl acetate 1.5% in 70% Ethanol and lead citrate and observed using a Tecnai F20 transmission electron microscope at 200KV in the CoMET MRI facilities, INM, Montpellier France.

Semithin (1 μm) cross section were stained with toluidine blue (Sigma-Aldrich, 89640-5G) and observed with a Nanozoomer Hamamatsu. The g-ratio was determined using the ImageJ GRatioCalculator plug-in (CIF, UNIL).

## Metabolomic

Sciatic nerves of 4 months mice were dissected and immediately frozen in liquid nitrogen until metabolomics analysis. Samples were homogenized with 2 grinding cycles, each at 6600 rpm for 20 sec, spaced by 20 sec, using a Precellys homogenizer (Bertin Technologies, Montigny-le-Bretonneux, France) kept in a room at +4˚C. The supernatant was recovered after centrifuging the homogenate and kept at -80˚C until mass spectrometric analysis. Targeted metabolomic analysis was carried out using the AbsoluteIDQ kit p 180 (Biocrates Life Sciences AG,

Innsbruck, Austria). This kit standardizes mass spectrometry quantification (Sciex QTRAP 5500 AB mass spectrometer, SCIEX, Villebon-sur-Yvette, France) of 188 metabolites including 40 acylcarnitines, 21 amino acids, 21 biogenic amines, 90 glycerophospholipids, 15 shingoli-pids and the sum of hexoses. The full list of individual metabolites is available at http://www. biocrates.com/products/research-products/absoluteidq-p180-kit. Flow injection analysis cou-pled with tandem mass spectrometry (FIA-MS/MS) was used for the analysis of carnitine, acyl-carnitines, lipids and hexoses. Liquid chromatography (LC) was used for separating amino acids and biogenic amines before quantitation with mass spectrometry. Before statistical analy-sis, the raw metabolomics data were examined to exclude metabolites with more than 20% of concentration values below the lower limit of quantitation (LLOQ) or above the upper limit of quantitation (ULOQ). Before performing statistical analyses, data from each sample were nor-malized by the total ion current (TIC).

To determine if molecule content variations reflected differences in metabolism in between Wild-Type, control and mutant mice, we carried out a Principal Component Analysis (PCA) by using the R-package ade4 [23]. This method is a classically used ordination method to sum-marize the patterns of variations among a large collection of samples, as it is well suited to con-tingency tables. As female's metabolism is more hormonal dependent than male, we excluded female mice to remove noise in the PCA. We thus performed the PCA on 15 male mice (5 WT, 5 controls and 5 mutants) with the full dataset of 161 molecules. Then we analyzed acyl-carnityls profiles in control mutant and mutant mice using a paired two-tailed T-tests.

## Lactate assay

To measure lactate content in sciatic nerves and forebrain of Control and Mutant animals, equal amounts of tissue were dissected and immediately homogenized with Lysing Matix D tubes (MP Biomedical) in the lactate assay buffer and proceed according the instruction of the lactate assay kit (MAK064, Sigma-Aldrich).The absorbance at 570 nm was measured with microplate reader (CLARIOstar, BMG Labtech)

## Western blot

Sciatic nerves were dissected from mice, after removal of the epineurium and perineurium, the nerves were frozen in liquid nitrogen and conserved at -80˚C. Protein were extracted from whole sciatic nerves after homogenization by sonication in standard RIPA lysis buffer and the protein concentration was measured using a BCA protein assay kit (Pierce). 20 μg proteins were directly analyzed by western blot using standard procedures with 12% SDS–PAGE and transferred on PVDF membranes for immunoblotting. Primary antibodies: Rabbit anti-Pkm1 1/2500 (NBP2-14833, Novus Biologicals); Rabbit anti-Pkm2 1/2500 (SAB4200095, Sigma Aldrich); Mouse anti-α-β-Actin 1/10000 (C1.AC-15, #A1978, Sigma Aldrich). Secondary anti-bodies, Peroxidase Goat anti Rabbit or Mouse (H+L) (Jackson Immuno Research) were used at 1/10000 dilution.

## Semi-quantitative rtPCR

Whole sciatic nerves from 2 and 10 days old mice (P2 and P10) or endoneuria from 1 (P28), 3 and 12 months-old mice were dissected at indicated time-points. Tissues were lysed using Tri-zol reagent and mechanical homogenization (TissueLyser II, Qiagen). Total RNA was extracted with the RNeasy lipid tissue kit (Qiagen), and RNA quality and concentration were verified by ND-1000 spectrophotometer (NanoDrop). cDNA was synthesized using 70-200ng of the RNA with the PrimeScript RT kit (Takara), following manufacturer's protocols. PKM1 and PKM2 splicing assays at different developmental time-points were conducted as

previously described [24] using the following primers PKM-F: 5′-ATGCTGGAGAGCATGAT-CAAGAAGCCACGC-3′ and PKM-R: 5′-CAACATCCATGGCCAAGTT-3′ with PCR annealing step at 60°C during 35 cycles and using HotStart Taq Polymerase (Qiagen). Digestions of cleaned 502bp-PCR products (with DNA Extraction kit, Qiagen) by NcoI (NEB), producing approx. 250 bp bands revealing PKM1 transcript, and PstI (NEB), producing approx. 280+220 bp bands revealing PKM2 transcript), were performed for 2h at 37°C. Digested PCR products were resolved on a 2% agarose gel, imaged by ChemiDoc XRS+ system and quantified using Image lab software version 3.0 (Biorad). The ratio between detected intensity of uncut and cut products was used to assess relative amount of PKM1 and PKM2.

## Probes construction and AAV preparation

pAAV-Laconic and pAAV-δGlu6 were obtained by digesting Laconic/pcDNA3.1(-) (gift from Luis Felipe Barros; Addgene plasmid #44238) [25] and pcDNA3.1 FLII12Pglu-700uδ6 (gift from Wolf Frommer; Addgene plasmid # 17866) [26] by BamH1/HindIII (NEB) and cloned into pAAV-MCS (Cell Biolabs, Inc.) under of a CMV promoter. Clones were validated by sequencing. pcDNA-mito-AT1.03 (from H. Imamura, Tokyo, Japan) [27] was digested with XhoI/HindIII (NEB), blunted and cloned into the CMV promoter controlled pAAV-MCS. Mitochondria-targeting tags were two tandem copies of CoxVIII. pAAV9 were produced at UPV, Universitat Autonoma de Barcelona or at INSERM U1089 University of Nantes, France.

## *In vivo* injection in spinal cord

The pups between 1 and 3 days were covered in aluminum foil and completely surrounded in ice for 3–4 min, until a completely cryoanesthetizia. Cryoanesthetized neonates were injected using a thin glass needle filled with colored viral and hold by a micromanipulator (IM—3C, Narishige Japan Group) directly in the spinal cord. 1 μL of the viral solution was injected slowly with short pressure pulses using a microinjector (Pneumatic Picopump PV820, World Precision Instruments) coupled to a 3 MHz function pulse generator (GFG8215, Langlois). The injection site was then cleaned with betadine (Vetoquinol, cat. No. 3042413) and pups were warmed in hands. When fully awake, pups were put back to the mother and the littermates.

## Imaging and stimulation of saphenous nerves in living mice

3–4 weeks after the viral injection, mice were anesthetized with a constant low (1.5 l/min) of oxygen+ 5% of isoflurane in an anesthesia induction box (World precision Instrument, Ref. EZ-B800) for 5 min. Thereafter the anesthesia was maintained with a mask delivering 2% isoflurane at 0.8 L/min. The eyes were protected with Ocry-gel (TVM, cat. No. 48026T613/3). The mouse was placed on the back in a silicone mold, the hind leg was shaved and the paws were immobilized using small pins. The incision area was disinfected with betadine (Vetoquinol, cat. No. 3042413). After incision of the skin of the thigh, connective tissue was carefully removed to expose the saphenous nerve. For electrophysiological experiments, the saphenous nerve was lifted up in the middle of the thigh and isolated electrically from the surrounding muscle tissue using a plastic trip. A pair of stimulating platinum electrodes (World precision Instrument, PTM23B05KT) held by micromanipulators (U-31CF, Narishige) was carefully placed under the saphenous nerve, at the ankle. The recording electrode (AD Instruments, MLA 1203) was placed at the groin. A reference needle electrode was inserted in the groin area and a ground electrode is placed in the tail. Once the electrodes were positioned the animal was transferred in a chamber at 37°C under the two-photon microscope (LSM 7 MP OPO, Zeiss), electrodes are connected to a Powerlab 26T (AD Instrument; ML4856). Supramaximal

stimulus (between 4 and 12 mA, 150 μs) was delivered at a frequency of 10 Hz during 5 min [28]. Time-lapse of the middle portion of the exposed nerve in the thigh were acquired every 5 min during 1 h with 20X objective lens (LD CApochromat, 421887–9970, Zeiss. For Laconic, δGlu6, and mito-ATeam alike, a 870 nm excitation wavelength was used to obtain both emission wavelength 475 nm for mTFP (Laconic) or CFP (δGlu6, mito-Ateam) and emission wavelength 527 nm for Venus (Laconic, mito-ATeam) or YFP (δGlu6).

### Image analysis

We used ImageJ software to analyze the relative lactate, glucose or mitochondrial ATP levels in peripheral axons. The acquired images for each wavelength (em. 475 nm for mTFP or CFP and em. 527 nm for Venus or YFP) were aligned using the Template Matching plugin. We defined a Region of Interest (ROI) encompassing the cytosolic area of one axon or encompassing all labelled mitochondria of one axon and the mean fluorescent intensity in the ROI was measured on both images. These light intensities were then corrected for background light intensity determined as an area within the nerve where no fluorescent signal from the viral probe can be observed. The Citrine/CFP (δGlu6) or Venus/mTFP (Laconic) or Venus/CFP (mito-Ateam) ratio was then calculated from the 2 values for mean light intensity.

The MTrackJ plugin in ImageJ was used for analysis of mitochondrial movement. For at least 5 images over a time span of 20 minutes, the location of the same mitochondrion was marked, the distance between marked locations was measured and migration velocity was calculated. n = 65 (Control before stimulation), 75 (Mutant before stimulation), 76 (Control after stimulation), 45 (Mutant after stimulation) mitochondria.

### Ex vivo nerve electrophysiology

Sciatic nerves were dissected out and transferred into oxygenated artificial cerebrospinal fluid (ACSF) containing 126 mM NaCl, 3 mM KCl, 2 mM $CaCl_2$, 2 mM $MgSO_4$, 1.25 mM $NaH_2PO_4$, 26 mM $NaHCO_3$, and 10 mM dextrose, pH 7.4–7.5. The nerves were desheated, and cut into 2 cm segments. Nerves were then placed in a three compartment recording chamber and perfused (1–2 ml/min) in 36°C ACSF equilibrated with 95% $O_2$-5% $CO_2$. The distal end was stimulated supramaximally (40 μs duration) through two electrodes isolated with Vaseline, and recordings were performed at the proximal end. Signals were amplified and digitized at 500 kHz. Measurements were made once the effects had reached a steady state. The delay and duration of compound action potentials (CAPs) were calculated at half the maximal amplitude and at the maximal amplitude. For recruitment analysis, the amplitude of CAPs was measured and plotted as a function of the stimulation intensity. For refractory period analysis, two stimuli were applied at different intervals, and the amplitude of the second CAP was measured and plotted as a function of the stimulus interval. To ensure that the amplitude of the second response was accurately assessed, the first response was subtracted from all the recordings. For train stimulation, nerves were stimulated for 200ms at frequencies ranging from 100 to 1 kHz. The amplitude of each evoked CAP was measured and plotted as a function of time. Conduction velocities were estimated from latencies.

### Statistical analysis

Except indicated otherwise, statistical analysis were done using Graphpad Prism (GraphPad Software, San Diego, USA). Statistical significance: * = P value<0.05; ** = P value<0.01; *** = P value<0.001.

## Results

### PKM2 expression promotes aerobic glycolysis in mSC

Firstly, to investigate the metabolic status of SC, we examined the expression and localization of PKM1 and PKM2 isoforms in mouse sciatic nerves using RT-qPCR and immunohistochemistry. PKM1 mRNA was downregulated during sciatic nerve maturation from postnatal day (P) 2 to P28 while PKM2 expression increased at the same time (**Fig 1A**). At non mature ages of SC, P4 and P15, PKM1 and PKM2 were both expressed in mSC (characterized by E-cadherin staining) [29] that surrounded axons (characterized by 2H3 staining, **Fig 1B a, b, c, d**). When SC were mature and formed myelin, at P30 and 5 months, PKM2 replaced PKM1 in mSC (**Fig 1B e, f, g, h**), in particular in the perinuclear region (**S1 Fig in S1 File**), suggesting mSC enters the aerobic glycolysis metabolic mode when the myelinated fibers are mature.

### PKM2 deletion in mSC leads to a reorganization of the metabolism in the nerve

The detected expression of PKM2 in mature mSC suggested that they were producing lactate, which raised the possibility that they provided a trophic support to axons. To test this hypothesis, we first performed a time-restricted conditional deletion of PKM2 isoform in myelinating glia of mice. A mouse strain expressing floxed PKM2-specific exon 10 alleles [14] was crossed with a strain expressing the Cre recombinase under the inducible and myelinating cell-specific promoter PLP1-ERt [22]. Injecting tamoxifen in Cre positive/*PKM2*<sup>fl/fl</sup> (mutant) and Cre-negative/*PKM2*<sup>fl/fl</sup> (control) littermates at 1 month of age resulted in a significant decrease of PKM2 expression in sciatic nerves of mutant as detected by immunoblotting, quantitative RT-PCR and immunostaining (**Fig 1C–1E**). The same experiments indicated that PKM1 was re-expressed in PKM2-deleted mSC (**Fig 1D and 1E**), suggesting a compensatory mechanism to maintain energy production.

Next, we expressed a lactate-detecting fluorescent probe (Laconic, **S2 Fig in S1 File**) in mSC of the mouse sciatic nerve *in vivo* using an AAV9 vector [30]. Mutant and control mice expressing the probe were then anesthetized and their nerves imaged using a multi-photon microscope to measure the relative amount of lactate in mSC of living mice. Cells of mutant mice showed significantly less lactate than mSC of control mice (**Fig 1F**). In addition, biochemical measurements showed that this deficit of lactate in mSC resulted in a global reduction of the amount of lactate in mutant mice nerves compared to control ones (**Fig 1G**). At the opposite, while PKM2 is also expressed in CNS oligodendrocytes [31], the amount of lactate in the frontal part of the mouse brain was not altered in mutant versus control mice (**S3 Fig in S1 File**), suggesting that PKM2 deletion has a limited impact on the CNS metabolism and function.

To analyze in a broader way the influence of PKM2 deletion in mSC on the metabolic status of the nerve, we performed a targeted metabolomic screen on mutant and control mouse nerves. We observed a decrease of some acylcarnitines in mutant mice (**S4 Fig in S1 File**), without modification of the acylcarnitine/carnitine ratio (0.2707 vs 0.2739 in control and mutant respectively, P = 0.77 two–tailed Student T-test). These acylcarnitines are involved in the transport of fatty acids across mitochondrial membranes before their degradation in carnitine and acetyl-CoA [32]. Therefore, the maintenance of the ratio coupled to the decrease of acylcarnitines indicated a higher turnover, suggesting an increase of the mitochondrial activity. Moreover, as mitochondrial dysfunction following deletion of TFAM1 (Transcriptional Factor A Mitochondrial 1) in mSC of mice has been shown to dramatically increase acylcarnitines [33], our data suggested the opposite effect in mutant mice, i.e. the upregulation of mitochondrial activity.

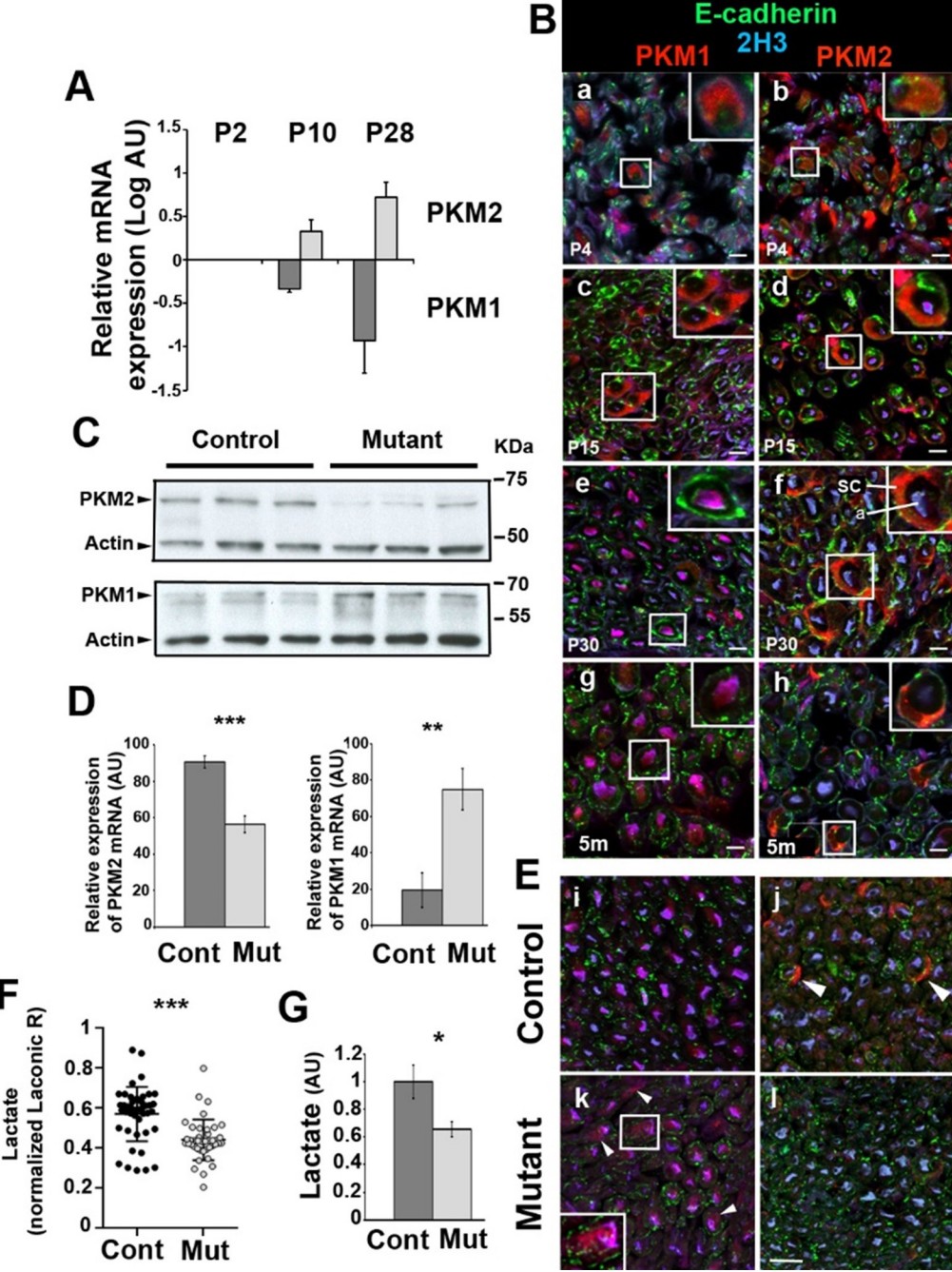

**Fig 1. PKM2 is efficiently deleted in mature mSC of mutant mice. A**- Quantitative RT-PCR on sciatic nerve extracts shows decreased PKM1 and increased PKM2 expressions during mSC maturation. Normalized on P2 values. n = 4 animals. **B**- Immunostaining on mouse sciatic nerve sections at increasing ages (a, b: 4 days postnatal (d); c, d: 15 days postnatal; e, f: 30 days postnatal; g, h: 5 months postnatal (m)) shows decreased PKM1 (left panels) and increased PKM2 (right panels) expressions in mSC over time. m: months. Sc: Schwann cell; a: axon. **C**- Western blots on sciatic nerve extracts show lower PKM2 and higher PKM1 amounts in Mutant versus Control mice sciatic nerves (2 months old). Actin serves as loading control. n = 3 mice. **D**- Quantitative RT-PCR on sciatic nerve extracts shows decreased PKM2 and increased PKM1 expressions in Mutant (Mut, n = 7) versus Control (Cont, n = 5) mouse sciatic nerves (samples from 3 and 12 months old mice were pooled). **E**- Immunostainings on sciatic nerve sections show PKM2 lower in Mutant mSC than in Control mice (panel j vs panel l) while PKM1 is higher (panel i vs panel k)(2 months old). **F**- The ratio of Laconic probe was measured in mSC of Mutant (n = 11 mice, 41 cells) and Control (n = 11 mice, 45 cells) anesthetized mice and normalized to the mean of Control values (7 to 10 weeks old mice). **G**-Biochemical measure of lactate shows a reduced concentration in Mutant versus Control sciatic nerves (n = 6 mice)(samples from 4 and 12

months old mice were pooled). All scale bars = 10μm. Two-tailed Student t-test. Error bars represent SD (A, D) or SEM (F). AU: arbitrary unit.

## Lactate homeostasis and ATP production are impaired in axons of PKM2-SCKO mice

According to the lactate shuttle theory, this decrease of lactate should also result in a shortage of axonal lactate. We investigated this assumption by expressing Laconic in peripheral axons using an AAV9 vector injected intrathecally in young pups (**S5A Fig in S1 File**). After tamoxifen induced recombination we performed live-imaging of probe-labeled axons that cross the saphenous nerve in anesthetized mice (**S5A Fig in S1 File**). The sensory saphenous nerve was chosen over more heavily myelinated nerves to prevent motor reflexes to alter imaging upon stimulation. While in resting conditions no difference could be seen between genotypes (**S6A Fig in S1 File**), when nerves were challenged with physiologically-relevant electrical stimulations to generate action potentials in type A fibers, axonal lactate increased shortly after the stimulation in control but dropped in mutant (**Fig 2A**). In the long term, control mice axons were able to maintain their lactate homeostasis while mutant mice axons could not (**Fig 2A**, red line).

Beside lactate, another way axons may feed their mitochondria is glucose-derived pyruvate. Thus, we used a glucose-specific probe (FLII12Pglu-700uδ6, **S5B Fig in S1 File**) [26] to investigate glucose level in axons in the same conditions. While in resting axons a similar amount of glucose was found in both genotypes (**S6B Fig in S1 File**), upon electrical stimulations glucose increased in axons of mutant mice while it remained stable in control mice (**2B Fig**), suggesting that axons of mutant mice mobilized glucose instead of lactate upon stimulations. We finally investigated the production of ATP by axonal mitochondria *in vivo* using a mitochondria-targeted fluorescent probe detecting ATP (ATeam, **S5C Fig in S1 File**) [27]. Again, no difference could be seen in resting conditions (**S6C Fig in S1 File**). However, when axons were stimulated mutant mice mitochondria failed to increase ATP production while this production increased in control mice (**Fig 2C**). Thus, in absence of aerobic glycolysis in mSC, lactate homeostasis and mitochondria ATP production are impaired in electrically active axons, despite the mobilization of glucose.

## Behavior impairment and neuromuscular junction loss in PKM2-SCKO mice are not due to demyelination

Since we observed a deficit in energy production in axons, we tested whether this had an impact on motor capacities of mutant mice. A longitudinal analysis revealed a significant deficit in mutant mice in both Rotarod and grip tests (**Fig 3A, 3B**), which primarily involve peripheral nerves functions, while in openfield test, which primarily involves brain functions, no defect was found (**S7 Fig in S1 File**). This suggested a direct influence of the metabolic changes observed in mutant mice axons on peripheral nerves functions. We next measured the nerve conduction velocity of sciatic nerves, and noticed it was not altered in mutant mice (**Fig 3C**) indicating that the myelin sheath was not affected. Accordingly, electron microscopy analyses showed correctly myelinated axons with a slightly increased g-ratio in mutant mice (**S8A and S8B Fig in S1 File**). Axonal diameter distribution was also slightly shifted toward larger axons in Mutant vs Control animals (**S8C Fig in S1 File**). As larger axons display a higher g-ration (S8A Fig in S1 File), this change may explain the g-ratio increase. However, the conduction of action potentials along myelinated axons was not altered as the electrophysiological properties of sciatic nerve axons *ex vivo* were maintained even at very high firing

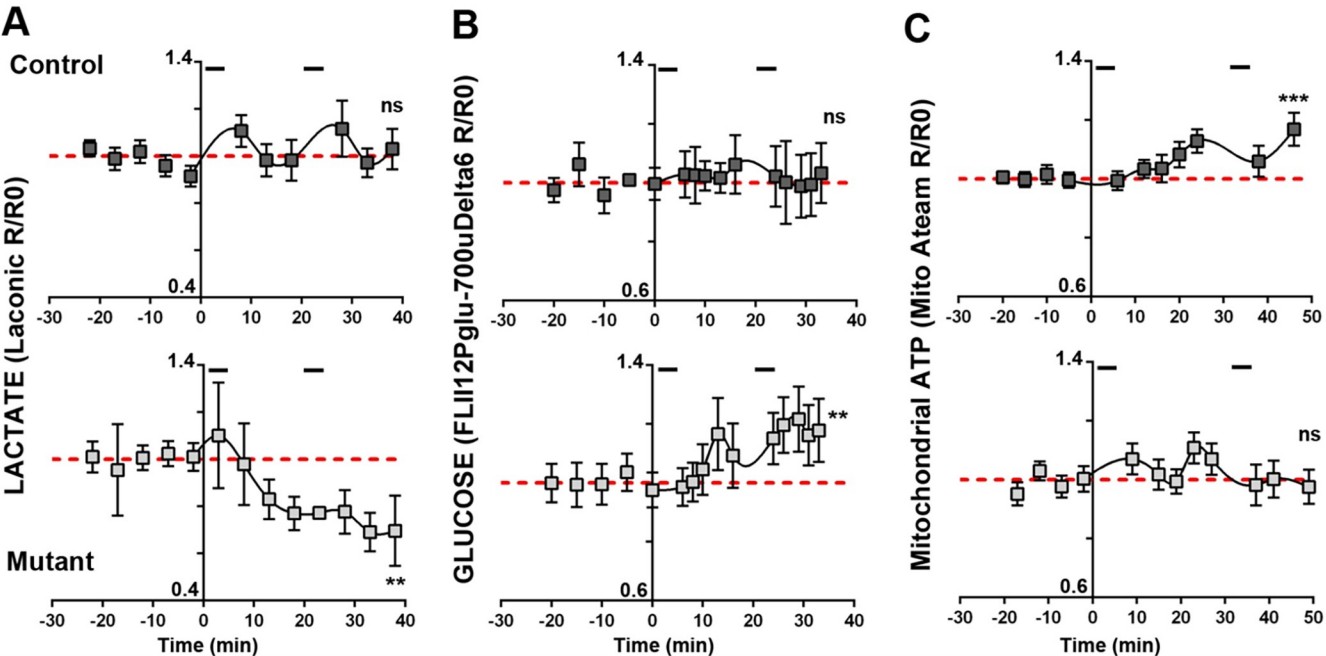

**Fig 2. Mutant mice axons display an altered metabolic response to electrical stimulations.** The ratio of fluorescent probes (R) was normalized over the average ratio at time points before the first stimulation (R0)(basal value). Segments at the top of each Control mice graphs show stimulation periods. Lines show the basal value line. Statistical analysis shows the test for linear trend following one-way ANOVA analysis. Error bars represent SEM. **A**- n = 20 axons, 7 mice for Control and 7 axons, 4 mice for Mutant. **B**- n = 6 axons, 3 mice for Control and 9 axons, 3 mice for Mutant. **C**- n = 16 axons, 5 mice for Control and 14 axons, 5 mice for Mutant.

frequencies (**S9 Fig in S1 File, S1 Table**). Motor neuron number did not change in the spinal cord (**Fig 3D**) but they displayed a higher expression of cleaved Caspase 3 (**Fig 3E** and **S9 Fig in S1 File**), a marker of neuronal stress and axonal degeneration [34]. Indeed, tracking axonal mitochondria *in vivo* revealed a decrease of their movements in mutant mice both before and after electrical stimulation (**Fig 3F**). In addition, mutant mice showed significantly more denervated neuromuscular junctions with intact postsynaptic structures (**Fig 3G and 3H**) in the gastrocnemius muscle, suggesting the retraction of axon terminals. Taken together, these data indicated that PKM2 deletion did not affect the maintenance of the myelin sheet but resulted in a motor distal neuropathy in mice. This indicated that aerobic glycolysis is required in mSC for the maintenance of axonal trafficking and neuromuscular junctions.

## Long-term maintenance of axons function relies on lactate feeding by SC

To investigate further the role of lactate intake in axons' long term maintenance, we treated mutant and control mice with dichloroacetate (DCA), a drug that promotes mitochondrial consumption of pyruvate and decreases the availability of lactate (**Fig 4A**) [20]. DCA treatment has been considered for patients suffering from congenital lactic acidosis and for cancer treatment but it can lead to a reversible peripheral neuropathy primarily affecting axons in humans and rodents [18]. Over the seven weeks of treatment, the chosen dose (500mg/kg daily) was not sufficient to induce the significant reduction of nerve conduction velocity (**Fig 4C**) reported previously at 1g/kg [35]. Nevertheless, Rotarod performance decreased in treated mice independently of the genotype (**Fig 4B**), suggesting an axonal defect. When the treatment stopped, control mice recovered over 3 weeks on the Rotarod while mutant mice did not (**Fig 4B**). This indicated that axonal function was definitely damaged in these mice, confirming that

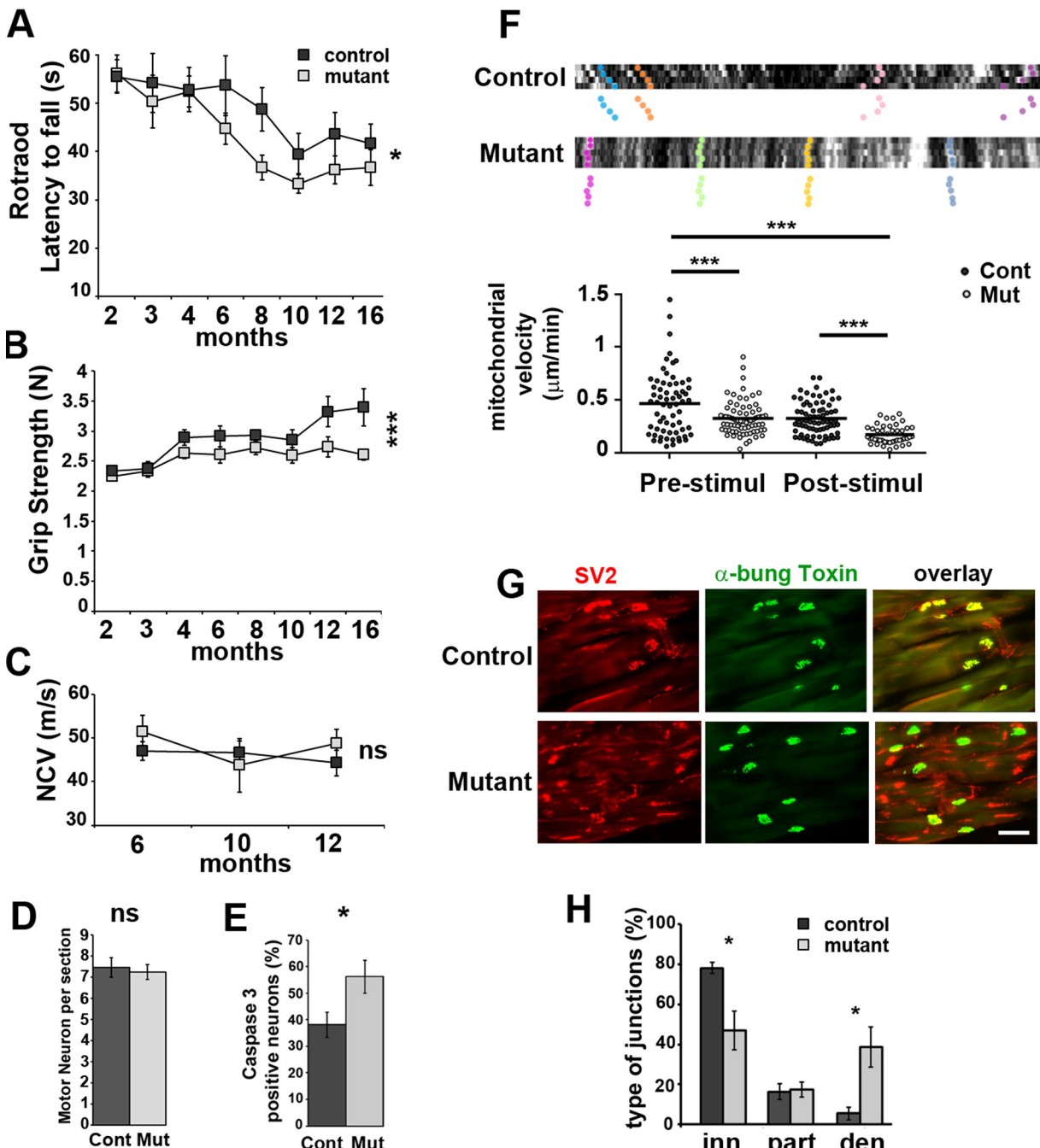

**Fig 3. Mutant mice show motor deficit and display axonal defects. A-** Rotarod latency to fall and **B-** grip strength were measured between 2 and 16 months postnatal on Control (n = 21) and Mutant (n = 25) mice. Two-way non repeated measures ANOVA Sidak's post-hoc tests. **C-** Nerve conduction velocity (NCV) measured in Control and Mutant mice (n = 10 each). Two-way non repeated measures ANOVA Sidak's post-hoc tests. **D-** Spinal cord sections of Mutant (Mut) and Control (Cont) mice (12 months) were stained with cresyl violet and motor neurons were counted (see S8 Fig in **S1 File**). n = 3 animals (14 to 30 sections). Two tailed Student t-test. **E-** Spinal cord sections of Mutant (Mut, n = 19 images, 3 animals) and Control (Cont, n = 12 images, 2 animals) mice were immunostained for Caspase 3, ChAT and Neurofilament (S8 Fig in **S1 File**) (12 months old). Caspase 3 positive neurons in percentage of Neurofilament positive neurons. Two tailed Student t-test. **F-** Axonal mitochondria labelled with mito-Dsred2 were imaged *in vivo* before and after electrical stimulation. **Upper panels** show typical kymographs. Tracked mitochondria are shown with colour dots. Mutant mice mitochondria follow a straight pattern indicating they are immobile or slowly moving. **Lower panel**: Mitochondria speed was plotted according to the genotype before and after stimulation. One-way ANOVA Tukey's post-hoc test. n are provided in Material and Methods. **G-** Gastrocnemius muscle neuromuscular junctions of 12 months old Mutant and Control mice were stained for presynaptic SV2 and postsynaptic acetylcholine receptor with FITC-α-bungarotoxin (α-bung). Scale bar = 100μm. **H-** Innervated (inn, complete overlap), partially innervated (part, partial overlap) and denervated (den, no overlap) junctions were counted on sections as shown in D. Two tailed Student t-test. n = 4 mice (12 months). Error bars represent SEM.

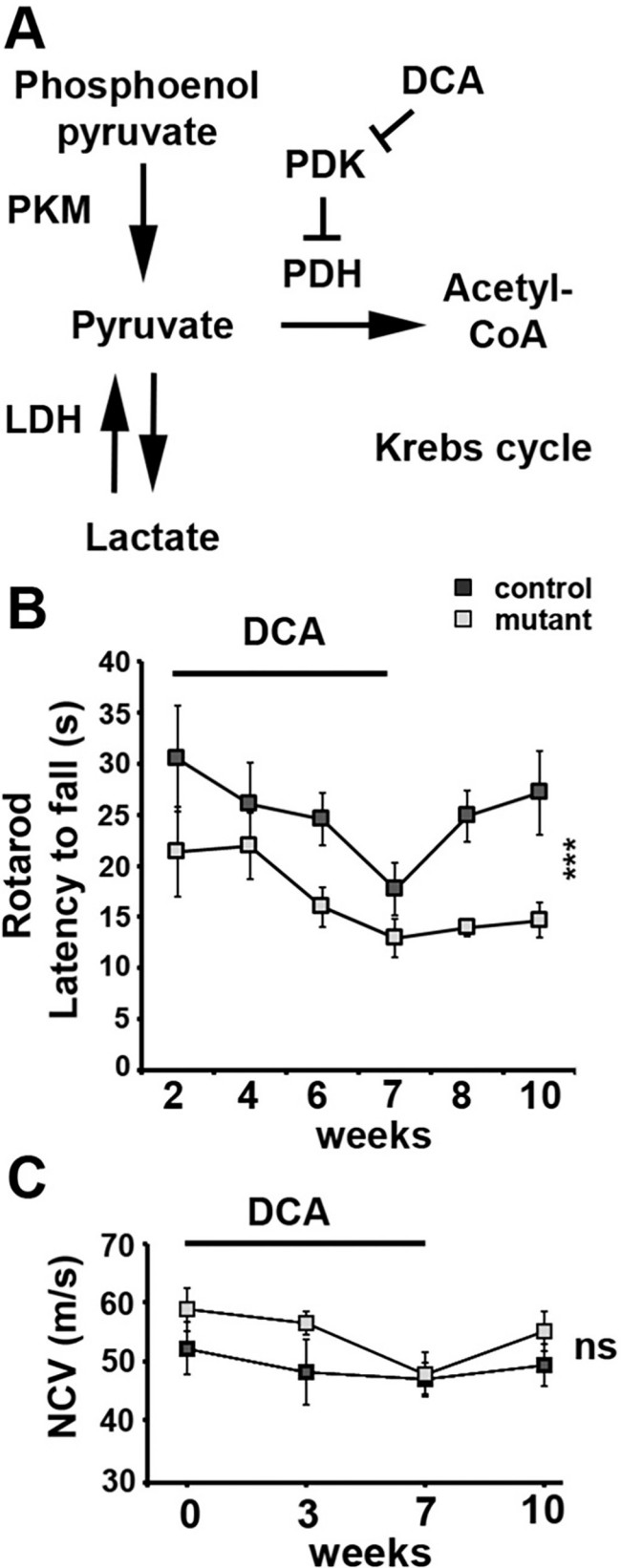

**Fig 4. PKM2 deletion in mSC impairs motor performance recovery after DCA treatment. A-** Metabolic pathways directly affected by the manipulations. LDH: lactate dehydrogenase; PDH: pyruvate dehydrogenase; PDK: PDH kinase. **B-** Rotarod latency to fall was measured in 9 months old Control (n = 5) and Mutant mice (n = 5) during a 7 weeks daily treatment with DCA and then during a three weeks recovery period without treatment. Two-way non repeated measures ANOVA Sidak's post-hoc test from week 7 to week 10. **C-** Nerve conduction velocity (NCV) measured in 9 months old Control (n = 5) and Mutant mice (n = 7) at 0, 3 and 7 weeks of DCA treatment and 3 weeks after stopping the treatment (week 10). Two-way non repeated measures ANOVA statistical tests, Sidak's post-hoc tests over the 4 time points. Error bars represent SEM.

aerobic glycolysis is required in mSC for the long-term maintenance of peripheral axons' physiology.

## Discussion

While the role of the lactate in the metabolic crosstalk between axons and the surrounding glia has been recognized for a long time in the CNS [1], its role remains unclear in the PNS. The presence of lactate in peripheral nerves had been detected and *ex vivo* experiments had shown that it could support action potentials propagation along axons [36]. However, the cellular origin of the released lactate and its relevance as an energy substrate for axons remains largely unclear [37]. By deleting PKM2 in mSC *in vivo* we showed that part of this nerve lactate is provided by mSC using aerobic glycolysis and this lactate is required for axons to maintain their metabolic homeostasis and functions over time.

CNS lactate and functions required for open field test were not affected by PKM2 deletion in myelin-forming cells. It remains possible that other CNS functions not involved in open field test are affected by PKM2 deletion and additional tests should be performed to investigate this. However, it is important to note here that CNS neurons are directly surrounded by at least two types of glial cells, astrocytes and oligodendrocytes, which are both able to shuttle lactate into neurons [6–8]. Thus, deleting PKM2 in only one of these glial cell types may not be sufficient to decrease brain lactate or to impair neuronal function.

Lactate is abundant in the body and in particular in the nervous system but axons appear unable to use this lactate to maintain their homeostasis. In these conditions, rescuing the phenotype of mutant mice through a supplementation in lactate appeared useless. However, allowing SC to produce more lactate despite the loss of PKM2 or specifically increasing lactate uptake by axons may represent a relevant but challenging strategy to rescue this phenotype.

Unexpectedly, and to the opposite of cancer cells, preventing mSC performing aerobic glycolysis had very little impact on its own physiology and biology. Indeed, no major defect could be detected in the myelin sheath of PKM2 mutant mice and the nerve conduction was not affected. A slight but significant shift occurred in the amount of some acylcarnitines without any alteration of the carnitine/acylcarnitine ratio, suggesting a higher turnover of these compounds that shuttle lipids into mitochondria for β-oxidation. According to the literature, we interpreted these data as the sign of a higher metabolic activity of mSC mitochondria. This is consistent with the upregulation of PKM1 expression we observed in mutant mice nerves as the higher enzymatic activity of this isoform promotes pyruvate production and its use in the mitochondrial citric acid cycle and respiration. In the absence of PKM2, mSC metabolism probably shifted to more oxidative phosphorylation. As mSC cannot be dispensed of mitochondrial respiration [7, 9], this shift is likely to be mild for the cell which is still able to produce myelin and maintain it. However, as evidenced by our results the shutdown of aerobic glycolysis in mSC is clearly deleterious for axonal maintenance. Therefore, we conclude that mSC use simultaneously both mitochondrial respiration for myelin production and maintenance and aerobic glycolysis to support axons survival.

The challenge in the analysis of metabolic crosstalk in a tissue is to distinguish between the metabolisms of the different cell types. In addition, nerve injury or section immediately modifies both axonal and glial mitochondria physiology [38]. To overcome these challenges, we chose to use the genetically encoded fluorescent probes delivered specifically to axons or mSC *in vivo* to follow a few critical metabolites in real time in physiological conditions. These probes such as lactate sensor *Laconic*, the glucose sensor *FLII12Pglu-700uδ6* and the ATP sensor *ATeam* have been extensively characterized in several *in vitro* and *in vivo* models and we confirmed that variations could be observed in the nerve *in vivo*. Although this gave us a limited view of all the metabolic changes that may have occurred in mutant cells, we could nevertheless draw some decisive conclusions.

Firstly, we observed that the metabolic homeostasis is really efficient in peripheral nerves. While electrically activated axons show radical alterations in mutant mice in just a few dozens of minutes in particular for lactate, no change could be detected in resting axons of anesthetized animals disregarding the genotype. In this regard, axonal homeostasis is much more efficient than mSC homeostasis as lactate levels remained significantly lower in these cells in mutant mice than in control mice. However, axonal homeostasis was not sufficient to buffer the metabolic changes that occur following physiological stimulations. Indeed, while axonal lactate remained apparently unaltered after stimulations in control mice, it sharply and steadily dropped in mutant mice axons. This revealed that actually the maintenance of lactate levels in control mice axons is due to a significant uptake of mSC lactate to compensate for a severe consumption during axonal activity [39]. Therefore, lactate produced by mSC is a critical fuel for mitochondria of actively-firing myelinated axons. Indeed, even the mobilization of glucose in stimulated axons was not sufficient to sustain an adequate ATP production in mitochondria. The involvement of the mSC lactate was definitively confirmed by the DCA treatment. Mutant mice were unable to recover their performances on the Rotarod following the draining of lactate supplies, while mice that could produce lactate in their mSC through aerobic glycolysis recovered. This underlines the dependence of myelinated axons to mSC lactate as all the other sources of energy substrates such as glucose, glycogen or glutamine are not directly altered by DCA. The reason for this dependence is puzzling but one possibility could be the swift accessibility of this glial lactate pool and its readiness for mitochondrial respiration while using glucose may require more time and enzymatic resources in the axons to generate pyruvate. This concept is supported by the recent discovery of a transient upregulation of glycolysis in mSC and the involvement of MCT to promote axons survival in an peripheral nerve injury model [40].

As we did not investigate the sensory function of peripheral nerves, the main macroscopic phenotype that we detected is a motor weakness observed through Rotarod and grip test in the absence of demyelination. This was not significantly detectable before 6 months suggesting a late onset of the distal motor neuropathy. Taken together, this is characteristic of axonal Charcot-Marie-Tooth diseases in mice. Moreover, this neuropathy did not result from motor neurons death but from an axonal dysfunction illustrated by a decreased mitochondrial motility and retracted neuromuscular terminals. This retraction is likely to be the main cause of the motor weakness. Mitochondrial migration, which is essential for the maintenance of the synapses, in particular in motor neurons [41], relies on the ATP produced in mitochondria. In a previous work we showed that shutting down ATP synthase activity steadily slowed down mitochondrial movements in mSC *in vivo* [42]. Taken together, our results suggest that, the failure of mutant mice axonal mitochondria to increase their ATP production following physiological activity is the likely initial cause of the axonal dysfunction.

However, axonal firing properties were not significantly altered by this mitochondrial failure. This was unexpected because a largely accepted idea is that axonal ATP, and therefore

axonal mitochondria, are required to maintain the membrane negative potential that allows depolarization. Actually, more recent analysis of neuronal energetics indicates that the energy cost of maintaining axon membrane potential is not that important, especially in myelinated fibers [1]. Therefore, the production of ATP by mutant mice axonal mitochondria may be sufficient to support this cost. Nevertheless, our data clearly indicate that mitochondrial ATP production and adaptation to the axon activity is critical for axonal maintenance.

Matching the energy demand of axons is critical to ensure the long-term maintenance of the nervous system. Indeed, increasing evidences indicate that an unbalanced energy supply to axons is a capital factor of neurodegenerative diseases such as ALS [43], Parkinson and Alzheimer diseases [44] or leprosis *Mycobacterium Leprae*-induced peripheral neuropathy [45]. However, the molecular basis of this support remains unclear. The present data show that axonal dysfunction may be a direct result of perturbed metabolic support by glial cells such as mSC. In this regard, the effect of DCA treatment on control and mutant mice motor performances is striking and the inability of mutant mice to recover from this treatment is challenging. Indeed, DCA is proposed as a long-term treatment for several cancers and other diseases such as acute and chronic lactic acidosis and diabetes. Our data suggest that drugs targeting aerobic glycolysis metabolism as a treatment for these diseases may have a singular and deleterious effect on the peripheral nervous system. Therefore, further and finer characterization of the axon/glia metabolic crosstalk is required in particular in neurodegenerative diseases.

## Supporting information

**S1 File.**
(DOCX)

**S1 Table. Electrophysiological characteristics of control and mutant mice.**
(DOCX)

**S1 Data.**
(XLSX)

**S1 Raw images.**
(PDF)

## Acknowledgments

We thank H.Boukhaddaoui, C.Sar, V.Baecker and the imaging facility MRI, the INM animal facility, L.Diouloufet and C. Cazevieille.

## Author Contributions

**Conceptualization:** Nicolas Tricaud.

**Data curation:** Juan Manuel Chao de la Barca.

**Formal analysis:** Marie Deck, Gerben Van Hameren, Jérôme Devaux, Jean-Jacques Médard, Juan Manuel Chao de la Barca, Nicolas Tricaud.

**Funding acquisition:** Roman Chrast, Nicolas Tricaud.

**Investigation:** Marie Deck, Gerben Van Hameren, Graham Campbell, Nathalie Bernard-Marissal, Jérôme Devaux, Jade Berthelot, Alise Lattard, Jean-Jacques Médard, Benoît Gautier, Sophie Guelfi, Scarlette Abbou, Patrice Quintana, Juan Manuel Chao de la Barca, Nicolas Tricaud.

**Methodology:** Patrice Quintana, Nicolas Tricaud.

**Project administration:** Nicolas Tricaud.

**Supervision:** Pascal Reynier, Guy Lenaers, Roman Chrast, Nicolas Tricaud.

**Validation:** Marie Deck, Gerben Van Hameren, Jérôme Devaux, Jean-Jacques Médard, Juan Manuel Chao de la Barca, Pascal Reynier, Roman Chrast, Nicolas Tricaud.

**Visualization:** Marie Deck, Gerben Van Hameren, Jérôme Devaux, Jean-Jacques Médard, Juan Manuel Chao de la Barca, Nicolas Tricaud.

**Writing – original draft:** Marie Deck, Gerben Van Hameren, Nathalie Bernard-Marissal, Jérôme Devaux, Jean-Jacques Médard, Roman Chrast, Nicolas Tricaud.

**Writing – review & editing:** Marie Deck, Gerben Van Hameren, Graham Campbell, Nathalie Bernard-Marissal, Jérôme Devaux, Jade Berthelot, Alise Lattard, Jean-Jacques Médard, Benoît Gautier, Sophie Guelfi, Scarlette Abbou, Patrice Quintana, Juan Manuel Chao de la Barca, Pascal Reynier, Guy Lenaers, Roman Chrast, Nicolas Tricaud.

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
