## [Decision Letter · Decision Letter 0]

12 Feb 2022

PONE-D-21-39340Physiology of PNS axons relies on glycolytic metabolism in myelinating Schwann cellsPLOS ONE

Dear Dr. Tricaud,

Thank you for submitting your manuscript to PLOS ONE. After careful consideration, we feel that it has merit but does not fully meet PLOS ONE’s publication criteria as it currently stands. Therefore, we invite you to submit a revised version of the manuscript that addresses the points raised during the review process.

This is indeed an interesting line of investigation, and one that will likely be well received to the general readership, especially those who focus on peripheral neuropathy and axon-glia interactions. However, the reviewers have some concerns that should be addressed.

In particular, all reviewers raise points related to how the data may be insufficient to fully warrant the conclusions drawn in the Discussion. Each of the reviewers has suggestions for how the Discussion and interpretation may need to be considered based on potential discrepancies in the results and the conclusions presented. Having seen the reviewers' comments, should the authors feel they can address these concerns in revision they are strongly encouraged to do so. However, if additional experiments (Rev 1: rescue with lactate administration) or inclusion of additional data (e.g. Rev 2: mean axon diameter in control vs. mutant) such has those suggested by the reviewers are feasible, they would certainly strengthen the study and its conclusions.

Finally, please be sure to look at the checklists that each reviewer has provided, and address any concerns raised there. Each reviewer has provided valuable feedback, both major and minor concerns, that will undoubtedly improve the final product.

We look forward to receiving your revised manuscript.

Kind regards,

David J Schulz

Academic Editor

PLOS ONE

Journal Requirements:

3. We noted in your submission details that a portion of your manuscript may have been presented or published elsewhere. (This manuscript was previously submitted to a different PLOS journal as either a presubmission inquiry or a full submission.

PLOS Biology PBIOLOGY-D-21-00914) 

Reviewers' comments:

Reviewer's Responses to Questions

**Comments to the Author**

1. Is the manuscript technically sound, and do the data support the conclusions?

Reviewer #1: Partly

Reviewer #2: Partly

Reviewer #3: Partly

2. Has the statistical analysis been performed appropriately and rigorously? 

Reviewer #1: Yes

Reviewer #2: Yes

Reviewer #3: I Don't Know

3. Have the authors made all data underlying the findings in their manuscript fully available?

Reviewer #1: Yes

Reviewer #2: Yes

Reviewer #3: Yes

4. Is the manuscript presented in an intelligible fashion and written in standard English?

Reviewer #1: Yes

Reviewer #2: Yes

Reviewer #3: Yes

5. Review Comments to the Author

Reviewer #1: The goal of this study was to investigate the importance of lactate produced by PKM2 in myelinating Schwann cells (mSC) in peripheral axons and myelinating Schwann cells. The highlight of this study was that it showed that aerobic glycolysis may be more important for the maintenance of axonal function than myelination. The evidence for the importance of aerobic glycolysis for axonal maintenance was that the loss of PKM2 only led to deficits in axonal function and not myelin deficits. I believe the authors’ interpretation of their findings was justified with relevant experiments accompanied by proper controls. The findings are interesting, and the thorough approaches were used to reveal the role of aerobic glycolysis in maintaining neuronal function via mSC. The authors also highlight the translational relevance of their findings to neurodegenerative diseases where axonal energy imbalance is a key feature. They also suggest that caution should be taken when treating patients with drugs that target aerobic glycolysis as these drugs could damage the peripheral nervous system.

The strengths of study include:

1) Characterizing the localization of PKM1 and 2 during early postnatal development was strong evidence that gave insights into the functional divergence between the PKM1 and PKM2.

2) There was good validation of the effectiveness of mouse PLP-PKM2 KO mouse model as various molecular techniques showed the downregulation of PKM2. There was also good proof of principle that reduction in PKM2 reduced lactate in mSC.

3) The use of specific probes for aerobic glycolytic substrates/ products allowed direct quantification in the changes of these substrates with the loss of PKM2

However, there are a few concerns need to be addressed by the authors before the work can be published.

Major concerns:

1) PLP1 is not only expressed in myelinating Schwann cells, but also in myelinating oligodendrocytes in the CNS. Additionally, the authors didn’t address whether PKM2 is mostly expressed in the PNS or has more relevance to the PNS. Therefore, the behavioral changes in the mice could also be a result of dysfunction in the CNS.

2) There were no rescue experiments presented. If the authors believe that reduction of lactate is the main reason causing neuronal dysfunction. Instead of further inhibiting lactate production with DCA treatment, they should supply the lactate to the animals and see whether that could rescue the phenotypes, because PMK2 KO could lead to many other defects (i.e. mitochondrial defects) reducing besides lactate production.

Minor Issues:

1) The RT-PCR results from figure 1D does not reflect protein expression level of figure 1C. Would be helpful to normalize the immunoblot data from 1C to help with the variation in the control.

2) There are several grammatical errors throughout the paper for example lines 51, 77, and 115.

3) In the abstract, it is said that PKM2 is upregulated in mSC, however, no comparison is given i.e. what condition is the upregulation compared to?

4) The authors mention abbreviations such as TFAM1 and CMT but these were never introduced to the reader.

5) It was not mentioned whether the electrical stimulations are physiologically relevant/ significant.

6) The authors need to justify why a PLP-cre mice was used instead of a PNS specific mouse model such as MPZ-cre.

7) In the summary it would be a good idea to explain briefly to the readers what the lactate shuttle theory is instead of assuming they know it. This will give the reader a better initial understanding as they begin to read the paper i.e. improve the readability.

8) In the summary, it would also be good to state that aerobic lactate production by PKM2 is important for neuron/ glia cells just as how the authors stated it is important for cancer.

9) The study is focused on mSC, the authors need to clearly state whether this focus is mainly because maintaining metabolic homeostasis is more important for neurons with long axons.

Reviewer #2: Deck and colleagues present a phenotyping of peripheral nerve structure and function in a mouse model of conditional pyruvate kinase M2 (PKM2) deletion ± superimposition of treatment with dichloroacetate (DCA). Despite expression of PKM2 in adult nerve being focused on Schwann cells, the authors build an argument that impaired lactate transfer to axons in PKM2 mutant mice results in a distal degenerative axonopathy.

The studies are novel, potentially important to understanding both basic neurobiology and the pathogenesis of peripheral neuropathies and use state of the art techniques. Comments:

1. Phenotyping of neuropathy is limited and lacks a number of widely used assessments that are standard for the field in rats or mice. The absence of behavioral, functional and structural assays for large and small sensory fibers is particularly troubling, given that sensory dysfunction is a common feature of many peripheral neuropathies and that large and small fiber sensory disorders are reported in the PNS of rats with DCA-induced neuropathy (see work by Stacpoole and colleagues, JNEN 2009).

2. The lack of MNCV slowing in DCA treated mice (Fig 4) is inconsistent with prior reports in rats (see above mentioned paper) and mice (Stacpoole et al Int. Rev. Neurobiol. 2019). This discrepancy should be discussed.

3. The focus on motor neuropathy in the discussion is also not consistent with the data generated. While some measures of motor neuropathy are reported (grip strength, loss of NMJ), characterization of PKM1/PKM2 expression is performed in the sciatic nerve, so does not distinguish between large motor and sensory fibers, while a number of other experiments, such as the studies of lactate and ATP homeostasis (Fig 2, Fig S5) are performed in saphenous nerve, a sensory nerve.

4. Given that the authors have pretty good EM images and have calculated G-ratio and axonal diameter, it would be very helpful to know if mean axonal diameter was different between the control and mutant mice (Fig S6), as this value is reduced in the sural (sensory) and tibial (mixed) nerves of rats with DCA neuropathy (see above mentioned paper). A plot of the size: frequency distributions for both groups would suffice.

5. References need to be provided in English (months are in French).

Some of the above can be addressed using images available to the authors. However, as written, the discussion is not entirely consistent with the data and develops a motor neuropathy focused agenda that, in the absence of sensory phenotyping, is unwarranted. Without requiring additional studies in new colonies of mice, the discussion should be revised to a) clearly identify data produced in motor vs sensory vs mixed nerves, b) de-emphasize the focus on motor neuropathy and c) acknowledge deficiencies in phenotyping that have excluded measures of sensory neuropathy.

Reviewer #3: Deck et al demonstrate that PKM2 is upregulated in myelinating Schwann cells (mSC) and deleting PKM2 in mice (through PLP expressing cells) (Schwann cells in the PNS) leads to reduced lactate in mSC and an axonal neuropathy in mice. The authors conclude that lactate production through aerobic glycolysis is necessary in mSC for the maintenance of axonal physiology in the PNS. The authors cite literature from the CNS in which astrocytes and oligodendrocytes metabolize glucose into lactate via the “lactate shuttle”. Depletion of PKM2 led to a deficit in Rotarod and grip strength. However ,nerve conducitons, CMAP amplitudes, and g-ratios morphologically were not changed. Mitochondrial movements were reduced and neuromuscular junctions appeared increasingly denervated. Treating mice with dicholoroacetate (DCA), which increases mitochondrial consumption of pyruvate and decreases lactate availability, exacerbated Rotarod dysfunction but did not affect the NCV. The authors surmise that DCA, a potential treatment for cancers and diabetes may constitute a risk for peripheral nerve function in patients.

The manuscript was well done and the biology is interesting. However there is concern about the significance of the abnormalities caused by knocking down PKM@. Chaanges in Rotaarod and grip without changes in neurophysiology or overall morphology (normal g-ratios) are always concerning. The data on partially innervated or denervated neuromuscular junctions is not convincing. The magnification is difficult to interpret on the immunohistochemistry. The gastrocnemius is a difficult muscle to quantitate NMJ changes on.

Finally raising a potential clinical concern for patients based on these abnormalities is probably not warrented

6. PLOS authors have the option to publish the peer review history of their article (what does this mean?). If published, this will include your full peer review and any attached files.

Reviewer #1: No

Reviewer #2: No

Reviewer #3: No

---

## [Author Response · Author response to Decision Letter 0]

8 Jun 2022

Reviewer #1: The goal of this study was to investigate the importance of lactate produced by PKM2 in myelinating Schwann cells (mSC) in peripheral axons and myelinating Schwann cells. The highlight of this study was that it showed that aerobic glycolysis may be more important for the maintenance of axonal function than myelination. The evidence for the importance of aerobic glycolysis for axonal maintenance was that the loss of PKM2 only led to deficits in axonal function and not myelin deficits. I believe the authors’ interpretation of their findings was justified with relevant experiments accompanied by proper controls. The findings are interesting, and the thorough approaches were used to reveal the role of aerobic glycolysis in maintaining neuronal function via mSC. The authors also highlight the translational relevance of their findings to neurodegenerative diseases where axonal energy imbalance is a key feature. They also suggest that caution should be taken when treating patients with drugs that target aerobic glycolysis as these drugs could damage the peripheral nervous system.

The strengths of study include:

1) Characterizing the localization of PKM1 and 2 during early postnatal development was strong evidence that gave insights into the functional divergence between the PKM1 and PKM2.

2) There was good validation of the effectiveness of mouse PLP-PKM2 KO mouse model as various molecular techniques showed the downregulation of PKM2. There was also good proof of principle that reduction in PKM2 reduced lactate in mSC.

3) The use of specific probes for aerobic glycolytic substrates/ products allowed direct quantification in the changes of these substrates with the loss of PKM2

However, there are a few concerns need to be addressed by the authors before the work can be published.

Major concerns:

1) PLP1 is not only expressed in myelinating Schwann cells, but also in myelinating oligodendrocytes in the CNS. Additionally, the authors didn’t address whether PKM2 is mostly expressed in the PNS or has more relevance to the PNS. Therefore, the behavioral changes in the mice could also be a result of dysfunction in the CNS.

PLP1 and PKM2 are indeed expressed in CNS oligodendrocytes, cells that play a role in the motor behaviour among other brain functions in mice. In another project we investigated the role of PKM2 in the CNS glia and we used PLP-ERT2-PKM2 flox mutant mice in an Open Field (OPF) experiment, a assay that typically address CNS defects. No deficiency was found in homozygous mice vs heterozygous mice using this assay (see figure below) suggesting that brain defects are absent or too mild to be detected using much simpler motor tests such as grip test and Rotarod. In addition, we did not find any decrease in lactate in the brain of mutant mice suggesting that deleting PKM2 in CNS oligodendrocytes has no impact on lactate supply to neurons.

We propose to add the lactate dosing data to the manuscript as supplementary Figure S3 (lactate dosing in mouse brain) and OPF data in Figure S7. These data are described in the manuscript page 17 and page 19 and discussed page 21 in yellow underlining.

2) There were no rescue experiments presented. If the authors believe that reduction of lactate is the main reason causing neuronal dysfunction. Instead of further inhibiting lactate production with DCA treatment, they should supply the lactate to the animals and see whether that could rescue the phenotypes, because PMK2 KO could lead to many other defects (i.e. mitochondrial defects) reducing besides lactate production.

Thank you for your comment. We agree that a rescue experiment would be a valuable addition to this project. However, based on considerations outlined below, we were unable to find an appropriate timely experiment to perform such a delicate rescue. 

The nervous system and nerves are full of lactate as is the blood. Nevertheless, the decrease of lactate production by SC induced by the glia-specific loss of PKM2 resulted in axonal defects among which loss of lactate homeostasis. So, even if the nerve is still full of lactate supplied by blood, axons are not able to use this lactate to maintain their homeostasis. Adding more lactate to the blood or to the nerve is therefore highly probably not going to resolve the problem that axons of mutant mice encounter. One potential approach to eventually rescue the phenotype would be by allowing SC to produce more lactate despite the loss of PKM2 or by increasing the uptake of lactate by axons themselves. While doable, these experiments go beyond supplying lactate to animals and will rather require to genetically modify axons or SC. 

We added a paragraph about this in the Discussion section (page 21).

Minor Issues:

1) The RT-PCR results from figure 1D does not reflect protein expression level of figure 1C. Would be helpful to normalize the immunoblot data from 1C to help with the variation in the control.

We have quantified our WB data and plotted them as seen below. PKM2/Actin and PKM1/Actin ratio significantly decreases and increases respectively accordingly to mRNA data. These data will be available in the “Raw data file” at the tab of Figure 1C.

 T-test Pvalue = 0.02

 T-test Pvalue= 0.12

2) There are several grammatical errors throughout the paper for example lines 51, 77, and 115.

We thank the reviewer for this insightful review and we corrected these spelling errors.

3) In the abstract, it is said that PKM2 is upregulated in mSC, however, no comparison is given i.e. what condition is the upregulation compared to?

Thank you for your comment. We modified the abstract as following “We show here that PKM2 is also upregulated in myelinating Schwann cells (mSC) of mature mouse sciatic nerve versus postnatal immature nerve.” 

4) The authors mention abbreviations such as TFAM1 and CMT but these were never introduced to the reader.

We improved the manuscript introducing these acronyms when they appear in the text. 

5) It was not mentioned whether the electrical stimulations are physiologically relevant/ significant.

We confirm that they were physiologically relevant, which we added in the Results section on page 18.

6) The authors need to justify why a PLP-cre mice was used instead of a PNS specific mouse model such as MPZ-cre.

“Plp1-creERT mouse line was chosen because it is the only commercially available line allowing selective and inducible recombination in myelinating Schwann cells.” was added to the Material and methods section.

7) In the summary it would be a good idea to explain briefly to the readers what the lactate shuttle theory is instead of assuming they know it. This will give the reader a better initial understanding as they begin to read the paper i.e. improve the readability.

We modified the last sentence of the summary as following “While lactate shuttle theory states that glial cells metabolize glucose into lactate to shuttle it to neurons, how glial cells support axonal metabolism and function remains unclear.”

8) In the summary, it would also be good to state that aerobic lactate production by PKM2 is important for neuron/ glia cells just as how the authors stated it is important for cancer.

We modified the last sentence of the summary as following “Thus, lactate production through PKM2 enzyme and aerobic glycolysis is essential in mSC for the long-term maintenance of peripheral nerve axon physiology and function.”

9) The study is focused on mSC, the authors need to clearly state whether this focus is mainly because maintaining metabolic homeostasis is more important for neurons with long axons.

In our first paragraph of the introduction we outlined the reason of our focus on peripheral axons and SC: “Energetic metabolism is an essential parameter of neurons’ function and survival (1–3). Indeed, the nervous system consumes a large amount of glucose mainly to allow synapses to function but also organelles and vesicles transport along axons and action potentials firing (1). In the peripheral nervous system (PNS), axons grow far from their cell bodies to reach distant targets such as muscles. Maintaining metabolic homeostasis on such a long distance is a real challenge for peripheral nerve axons (4). To support neurons in this challenge, myelinating Schwann cells (mSC) cover the large motor axons with a myelin sheath.”

Reviewer #2: Deck and colleagues present a phenotyping of peripheral nerve structure and function in a mouse model of conditional pyruvate kinase M2 (PKM2) deletion ± superimposition of treatment with dichloroacetate (DCA). Despite expression of PKM2 in adult nerve being focused on Schwann cells, the authors build an argument that impaired lactate transfer to axons in PKM2 mutant mice results in a distal degenerative axonopathy.

The studies are novel, potentially important to understanding both basic neurobiology and the pathogenesis of peripheral neuropathies and use state of the art techniques. Comments:

1. Phenotyping of neuropathy is limited and lacks a number of widely used assessments that are standard for the field in rats or mice. The absence of behavioral, functional and structural assays for large and small sensory fibers is particularly troubling, given that sensory dysfunction is a common feature of many peripheral neuropathies and that large and small fiber sensory disorders are reported in the PNS of rats with DCA-induced neuropathy (see work by Stacpoole and colleagues, JNEN 2009).

We agree with the reviewer that the phenotype we described in this mouse model is limited to the motor behaviour. In this regard, we performed two widely used tests to assess motor capacities of mice and rats: grip test and rotarod. As multiple sensory functions are mediated by non-myelinated fibers and as we modified myelinating SC only, we did not investigate the sensory function. A follow-up study will be necessary to investigate sensory functions in more detail.

2. The lack of MNCV slowing in DCA treated mice (Fig 4) is inconsistent with prior reports in rats (see above mentioned paper) and mice (Stacpoole et al Int. Rev. Neurobiol. 2019). This discrepancy should be discussed.

We modified the Discussion section on page 20 to expand on this discrepancy a bit further: “Over the seven weeks of treatment, the chosen dose (500mg/kg daily) was not sufficient to induce the significant reduction of nerve conduction velocity (Fig. 4C) reported previously (36,37), which suggests that nerve conduction is less dependent on mSC lactate in adult rats than in juvenile rats. Despite only a minor reduction in nerve conduction, Rotarod performance decreased in treated mice independently of the genotype (Fig. 4B), suggesting an axonal defect. When the treatment stopped, control mice recovered over 3 weeks on the Rotarod while mutant mice did not (Fig. 4B).”

3. The focus on motor neuropathy in the discussion is also not consistent with the data generated. While some measures of motor neuropathy are reported (grip strength, loss of NMJ), characterization of PKM1/PKM2 expression is performed in the sciatic nerve, so does not distinguish between large motor and sensory fibers, while a number of other experiments, such as the studies of lactate and ATP homeostasis (Fig 2, Fig S5) are performed in saphenous nerve, a sensory nerve.

The goal of the study was to investigate the role of PKM2 and lactate production by glycolytic pathway in myelinating SC on the axonal function. Therefore, we discriminated myelinated fibres vs non-myelinated fibres and not motor, sensory or motor and sensory nerves. As previously explained, we initially investigate motor behaviour because all motor fibres are myelinated while only a portion of the sensory fibres is myelinated. In addition, we investigated this portion of myelinated fibres in the sensory saphenous nerve in imaging experiments. The sensory saphenous nerve was chosen for imaging experiments to avoid motor reflexes upon stimulation, which could blur images. We added this clarification in the Results section on page 18.

4. Given that the authors have pretty good EM images and have calculated G-ratio and axonal diameter, it would be very helpful to know if mean axonal diameter was different between the control and mutant mice (Fig S6), as this value is reduced in the sural (sensory) and tibial (mixed) nerves of rats with DCA neuropathy (see above mentioned paper). A plot of the size: frequency distributions for both groups would suffice.

To answer the reviewer request, we re-analyzed our semi-thin EM pictures to collect data about much more fibres in both Mutant and Control animals. These data indicated a slight but significant increase in the g-ratio of Mutant mice (Figure S8B), suggesting that myelin sheath is slightly thinner in sciatic nerves of these mice. However, no evidence of demyelination or strong dysmyelination was seen as highlighted in the results section page 19.

These new data were also plotted as frequency distribution of axonal diameter in Figure S8C as requested by the reviewer. An unpaired two-tailed Student T-test was run over these data as described in the mentioned papers and no significant change was found. This result is described and highlighted in the results section on page 19.

5. References need to be provided in English (months are in French).

We corrected this.

Some of the above can be addressed using images available to the authors. However, as written, the discussion is not entirely consistent with the data and develops a motor neuropathy focused agenda that, in the absence of sensory phenotyping, is unwarranted. Without requiring additional studies in new colonies of mice, the discussion should be revised to a) clearly identify data produced in motor vs sensory vs mixed nerves, b) de-emphasize the focus on motor neuropathy and c) acknowledge deficiencies in phenotyping that have excluded measures of sensory neuropathy.

We acknowledged in the discussion that, in absence of investigation of the sensory function, our conclusions are limited to the motor function of the nerves as highlighted at page 23.

Reviewer #3: Deck et al demonstrate that PKM2 is upregulated in myelinating Schwann cells (mSC) and deleting PKM2 in mice (through PLP expressing cells) (Schwann cells in the PNS) leads to reduced lactate in mSC and an axonal neuropathy in mice. The authors conclude that lactate production through aerobic glycolysis is necessary in mSC for the maintenance of axonal physiology in the PNS. The authors cite literature from the CNS in which astrocytes and oligodendrocytes metabolize glucose into lactate via the “lactate shuttle”. Depletion of PKM2 led to a deficit in Rotarod and grip strength. However, nerve conductions, CMAP amplitudes, and g-ratios morphologically were not changed. Mitochondrial movements were reduced and neuromuscular junctions appeared increasingly denervated. Treating mice with dicholoroacetate (DCA), which increases mitochondrial consumption of pyruvate and decreases lactate availability, exacerbated Rotarod dysfunction but did not affect the NCV. The authors surmise that DCA, a potential treatment for cancers and diabetes may constitute a risk for peripheral nerve function in patients.

The manuscript was well done and the biology is interesting. However there is concern about the significance of the abnormalities caused by knocking down PKM@. Changes in Rotaarod and grip without changes in neurophysiology or overall morphology (normal g-ratios) are always concerning. 

We thank our reviewer for this comment. However, we would like to point out that hereditary peripheral neuropathies resulting from axonal dysfunction, Charcot-Marie-Tooth diseases type 2, are numerous and result in motor and sensory defects despite a normal nerve conduction velocity, normal g-ratio and limited alteration of the myelin sheath. 

The data on partially innervated or denervated neuromuscular junctions is not convincing. The magnification is difficult to interpret on the immunohistochemistry. The gastrocnemius is a difficult muscle to quantitate NMJ changes on.

Although challenging, our experiments were consistent with several previous studies based on immunohistochemistry and quantification of neuromuscular junctions in the gastrocnemius, even if this muscle is thick and may not be the most amenable to look at it. A similar study was recently published in PNAS (Bernard-Marissal N, van Hameren G, Juneja M, Pellegrino C, Louhivuori L, Bartesaghi L, Rochat C, El Mansour O, Médard JJ, Croisier M, Maclachlan C, Poirot O, Uhlén P, Timmerman V, Tricaud N, Schneider BL, Chrast R. Altered interplay between endoplasmic reticulum and mitochondria in Charcot-Marie-Tooth type 2A neuropathy. Proc Natl Acad Sci U S A. 2019 Feb 5;116(6):2328-2337. doi: 10.1073/pnas.1810932116. Epub 2019 Jan 18. PMID: 30659145; PMCID: PMC6369737).

Finally raising a potential clinical concern for patients based on these abnormalities is probably not warrented

Following reviewer’s suggestion, we removed the last sentence of the discussion.

---

## [Decision Letter · Decision Letter 1]

6 Jul 2022

PONE-D-21-39340R1Physiology of PNS axons relies on glycolytic metabolism in myelinating Schwann cellsPLOS ONE

Dear Dr. Tricaud,

Thank you for submitting your manuscript to PLOS ONE. After careful consideration, we feel that it has merit but does not fully meet PLOS ONE’s publication criteria as it currently stands. Therefore, we invite you to submit a revised version of the manuscript that addresses the points raised during the review process. All three reviewers agreed that you addressed all of their major concerns well. Reviewer 2 has a couple of minor suggestions for changes that will help make the results clearer and benefit the readership. If you could address these two points as well as you are able and resubmit, it would be appreciated. 

We look forward to receiving your revised manuscript.

Kind regards,

David J Schulz

Academic Editor

PLOS ONE

Journal Requirements:

Reviewers' comments:

Reviewer's Responses to Questions

**Comments to the Author**

1. If the authors have adequately addressed your comments raised in a previous round of review and you feel that this manuscript is now acceptable for publication, you may indicate that here to bypass the “Comments to the Author” section, enter your conflict of interest statement in the “Confidential to Editor” section, and submit your "Accept" recommendation.

Reviewer #1: All comments have been addressed

Reviewer #2: (No Response)

Reviewer #3: All comments have been addressed

2. Is the manuscript technically sound, and do the data support the conclusions?

Reviewer #1: Yes

Reviewer #2: Partly

Reviewer #3: Yes

3. Has the statistical analysis been performed appropriately and rigorously? 

Reviewer #1: Yes

Reviewer #2: No

Reviewer #3: Yes

4. Have the authors made all data underlying the findings in their manuscript fully available?

Reviewer #1: Yes

Reviewer #2: Yes

Reviewer #3: Yes

5. Is the manuscript presented in an intelligible fashion and written in standard English?

Reviewer #1: Yes

Reviewer #2: Yes

Reviewer #3: Yes

6. Review Comments to the Author

Reviewer #1: The authors have addressed all my concerns. I understand the difficulty of performing the lactate rescue experiments and including Mpz-cre.

Reviewer #2: The authors have made a good effort to address the concerns of the reviewers by amending text and adding new data and the manuscript is much improved. I have a couple of points that should be addressed.

1. In response to a request for comment on prior reports of MNCV slowing in DCA-exposed rodents the authors write “Over the seven weeks of treatment, the chosen dose (500mg/kg daily) was not sufficient to induce the significant reduction of nerve conduction velocity (Fig. 4C) reported previously (36,37), which suggests that nerve conduction is less dependent on mSC lactate in adult rats than in juvenile rats”. This interpretation is not correct as a) reference #37 reports MNCV slowing in adult DCA-exposed mice, not rats and b) reference #36 shows MNCV slowing in adult rats but not juvenile rats after exposure to 500mg/kg. DCA. The authors could simply note that their present MNCV data in adult mice exposed to 500mg/kg DCA does not replicate the prior report (ref #37) that used 1g/kg of DCA in adult mice – perhaps simply due to the lower dose used (note that 500mg/kg in rats as per reference #36 ≠ 500mg/kg in mice due absence of allometric scaling, whereas 500mg/kg in rats = 1g/kg in mice after allometric scaling).

2. The authors provide an interesting axonal size:frequency histogram (Fig S8C) and indicate that there was no difference between control and DCA exposed rats using an unpaired t test. I do not understand this analysis – what data were compared? The paper cited in the response to review used unpaired t test to compare mean axonal diameter, not the size:frequency histogram. Please report values (mean±SD) for MAD in the 2 groups to support any statistical analysis. It is also necessary to report fibers/nerve in order to fully interpret MAD or fiber size:frequency distributions, not just total number of fibers counted. Given that the size:frequency histogram (Figure S8C) suggests that DCA-exposed nerve may have a shift towards increased frequency of larger axons and reduced frequency of smaller axons, to what extent does increased axonal MAD contribute to the reported increase in g-ratio rather than the myelin thinning that the authors infer?

Reviewer #3: The authors have addressed my concerns. I have no additional concerns about the research ethics or publication ethics. This is an interesting study which adds knowledge to the field

7. PLOS authors have the option to publish the peer review history of their article (what does this mean?). If published, this will include your full peer review and any attached files.

Reviewer #1: **Yes: **Jian Hu

Reviewer #2: No

Reviewer #3: **Yes: **Michael E Shy

---

## [Author Response · Author response to Decision Letter 1]

10 Jul 2022

Reviewer #1: The authors have addressed all my concerns. I understand the difficulty of performing the lactate rescue experiments and including Mpz-cre.

Reviewer #2: The authors have made a good effort to address the concerns of the reviewers by amending text and adding new data and the manuscript is much improved. I have a couple of points that should be addressed.

1. In response to a request for comment on prior reports of MNCV slowing in DCA-exposed rodents the authors write “Over the seven weeks of treatment, the chosen dose (500mg/kg daily) was not sufficient to induce the significant reduction of nerve conduction velocity (Fig. 4C) reported previously (36,37), which suggests that nerve conduction is less dependent on mSC lactate in adult rats than in juvenile rats”. This interpretation is not correct as a) reference #37 reports MNCV slowing in adult DCA-exposed mice, not rats and b) reference #36 shows MNCV slowing in adult rats but not juvenile rats after exposure to 500mg/kg. DCA. The authors could simply note that their present MNCV data in adult mice exposed to 500mg/kg DCA does not replicate the prior report (ref #37) that used 1g/kg of DCA in adult mice – perhaps simply due to the lower dose used (note that 500mg/kg in rats as per reference #36 ≠ 500mg/kg in mice due absence of allometric scaling, whereas 500mg/kg in rats = 1g/kg in mice after allometric scaling).

We agree with the reviewer and we modified the results section page 20 line 430 as following: “Over the seven weeks of treatment, the chosen dose (500mg/kg daily) was not sufficient to induce the significant reduction of nerve conduction velocity (Fig. 4C) reported previously at 1g/kg (36). Nevertheless, Rotarod performance decreased in treated mice independently of the genotype (Fig. 4B), suggesting an axonal defect.”

2. The authors provide an interesting axonal size:frequency histogram (Fig S8C) and indicate that there was no difference between control and DCA exposed rats using an unpaired t test. I do not understand this analysis – what data were compared? The paper cited in the response to review used unpaired t test to compare mean axonal diameter, not the size:frequency histogram. Please report values (mean±SD) for MAD in the 2 groups to support any statistical analysis. It is also necessary to report fibers/nerve in order to fully interpret MAD or fiber size:frequency distributions, not just total number of fibers counted. 

Raw data including data relative to results presented in Figure S8C are shown in the “raw data” file. We agree with the reviewer that the Student T-test comparing the overall mean of axonal diameter is not accurate. We propose to use a two-way ANOVA test comparing genotypes in all axonal diameter category with a multiple analysis in each category. This analysis was done with Graphpad Prism and is presented in detail in the raw data file for Figure S8C. 

We found that, while no significant difference can be found between genotypes regarding each axonal diameter category, there is an overall significant interaction between axonal diameter and genotype. This was noted in the Figure S8C legend : “ Statistical two-way ANOVA analysis followed by a Sidak multiple comparison test comparing mean value for each axonal diameter category showed no significant difference between Control and Mutant mice (genotype P-value = 0.95; see raw data file). However, an overall significant interaction was found between genotype and axonal diameter category (interaction P-value = 0.017; see raw data file).” In addition, we noted this in the results section of the manuscript page 19 line 410: “Axonal diameter distribution was also slightly shifted toward larger axons in Mutant vs Control animals (Fig. S8C). As larger axons display a higher g-ration (Fig. S8A), this change may explain the g-ratio increase. However, the conduction of action potentials along myelinated axons was not altered as the electrophysiological properties of sciatic nerve axons ex vivo were maintained even at very high firing frequencies (Fig. S9, Table S).”

A comment about the statistical analysis was added to the Material and Methods section page 15 line 325: “Statistical analysis Except indicated otherwise, statistical analysis were done using Graphpad Prism (GraphPad Software, San Diego, USA). Statistical significance: *= P value<0.05; **= P value<0.01; ***= P value<0.001.”

Given that the size:frequency histogram (Figure S8C) suggests that DCA-exposed nerve may have a shift towards increased frequency of larger axons and reduced frequency of smaller axons, to what extent does increased axonal MAD contribute to the reported increase in g-ratio rather than the myelin thinning that the authors infer?

We indeed suggested that the decrease in the g-ratio may result from the shift of axon diameter distribution toward larger axons and therefore higher g-ratio.

Reviewer #3: The authors have addressed my concerns. I have no additional concerns about the research ethics or publication ethics. This is an interesting study which adds knowledge to the field

---

## [Editor Report · Decision Letter 2]

13 Jul 2022

Physiology of PNS axons relies on glycolytic metabolism in myelinating Schwann cells

PONE-D-21-39340R2

Dear Dr. Tricaud,

We’re pleased to inform you that your manuscript has been judged scientifically suitable for publication and will be formally accepted for publication once it meets all outstanding technical requirements.

Kind regards,

David J Schulz

Academic Editor

PLOS ONE
---

## [Editor Report · Acceptance letter]

23 Sep 2022

PONE-D-21-39340R2 

Physiology of PNS axons relies on glycolytic metabolism in myelinating Schwann cells. 

Dear Dr. Tricaud:

I'm pleased to inform you that your manuscript has been deemed suitable for publication in PLOS ONE. Congratulations! Your manuscript is now with our production department. 

Kind regards, 

on behalf of

Dr. David J Schulz 

Academic Editor

PLOS ONE